In situ growth and bioerosion rates of Lophelia pertusa in a Norwegian fjord and open shelf cold-water coral habitat

Büscher Janina V. 1 jbuescher@geomar.de
Wisshak Max 2
Form Armin U. 1
http://orcid.org/0000-0001-9373-9688 Titschack Jürgen 2 3
Nachtigall Kerstin 1
http://orcid.org/0000-0002-9442-452X Riebesell Ulf 1
1 Biological Oceanography, GEOMAR Helmholtz Centre for Ocean Research Kiel , Kiel , Germany
2 Marine Research Department, Senckenberg am Meer , Wilhelmshaven , Germany
3 Current Affiliation: Marine Sedimentology, MARUM—Center of Marine Environmental Sciences , Bremen , Germany
Benfield Mark
Electronic publication date: 2019 Sep 24
Publication date: 2019
Volume: 7
Electronic Location ID: e7586
Received 2019 Feb 26; Accepted 2019 Jul 30
Copyright: © 2019 Büscher et al.
Copyright year: 2019
Copyright holder: Büscher et al.
License: This is an open access article distributed under the terms of the Creative Commons Attribution License, which permits unrestricted use, distribution, reproduction and adaptation in any medium and for any purpose provided that it is properly attributed. For attribution, the original author(s), title, publication source (PeerJ) and either DOI or URL of the article must be cited.
License URL: https://creativecommons.org/licenses/by/4.0/

Keywords: Lophelia pertusa, Cold-Water Corals, In situ study, Growth, Bioerosion, North Atlantic, Inshore vs. offshore, Mortality

Funding: German coordinated project Biological Impacts of Ocean Acidification BIOACID II FKZ 03F0655A Federal Ministry of Education and Research (BMBF) This study was carried out as part of the German coordinated project Biological Impacts of Ocean Acidification (BIOACID II, Grant number: FKZ 03F0655A) funded by the Federal Ministry of Education and Research (BMBF). The funders had no role in study design, data collection and analysis, decision to publish, or preparation of the manuscript.

==============================
Coral reef resilience depends on the balance between carbonate precipitation, leading to reef growth, and carbonate degradation, for example, through bioerosion. Changes in environmental conditions are likely to affect the two processes differently, thereby shifting the balance between reef growth and degradation. In cold-water corals estimates of accretion-erosion processes in their natural habitat are scarce and solely live coral growth rates were studied with regard to future environmental changes in the laboratory so far, limiting our ability to assess the potential of cold-water coral reef ecosystems to cope with environmental changes. In the present study, growth rates of the two predominant colour morphotypes of live Lophelia pertusa as well as bioerosion rates of dead coral framework were assessed in different environmental settings in Norwegian cold-water coral reefs in a 1-year in situ experiment. Net growth (in weight gain and linear extension) of live L. pertusa was in the lower range of previous estimates and did not significantly differ between inshore (fjord) and offshore (open shelf) habitats. However, slightly higher net growth rates were obtained inshore. Bioerosion rates were significantly higher on-reef in the fjord compared to off-reef deployments in- and offshore. Besides, on-reef coral fragments yielded a broader range of individual growth and bioerosion rates, indicating higher turnover in live reef structures than off-reef with regard to accretion–bioerosion processes. Moreover, if the higher variation in growth rates represents a greater variance in (genetic) adaptations to natural environmental variability in the fjord, inshore reefs could possibly benefit under future ocean change compared to offshore reefs. Although not significantly different due to high variances between replicates, growth rates of orange branches were consistently higher at all sites, while mortality was statistically significantly lower, potentially indicating higher stress-resistance than the less pigmented white phenotype. Comparing the here measured rates of net accretion of live corals (regardless of colour morphotype) with net erosion of dead coral framework gives a first estimate of the dimensions of both processes in natural cold-water coral habitats, indicating that calcium carbonate loss through bioerosion amounts to one fifth to one sixth of the production rates by coral calcification (disregarding accretion processes of other organisms and proportion of live and dead coral framework in a reef). With regard to likely accelerating bioerosion and diminishing growth rates of corals under ocean acidification, the balance of reef accretion and degradation may be shifted towards higher biogenic dissolution in the future.

Introduction

Cold-water corals are important carbonate factories in the upper bathyal realm, which can build large reefs on continental shelves and slopes. Mean Holocene carbonate accumulation accounts to 103 g cm−2 kyr−1 in Norwegian cold-water coral reefs in the North Atlantic, therewith representing significant carbonate sinks on a local and potentially even global scale (Lindberg & Mienert, 2005; Titschack et al., 2015). Moreover, they are among the most prominent ecosystem engineers on Earth, hosting more than 2,700 species associated to cold-water coral reefs around the world, using them as nursery grounds or feeding places (Freiwald et al., 2004; Roberts & Cairns, 2014). In contrast to their tropical counterparts, cold-water coral reefs are usually built by only one or two dominant coral species. The most abundant framework-forming cold-water coral is the caryophyllid scleractinian Lophelia pertusa (Linnaeus, 1758). L. pertusa is widely distributed and builds extended reefs in almost all oceans except for the polar regions (Cairns, 1994; Rogers, 1999; Freiwald et al., 2004). Reefs built by L. pertusa have most frequently been found in the eastern Atlantic Ocean with a dense band of reefs extending from northern Norway in the Barents Sea to the coasts of West Africa (Freiwald et al., 2004). While this is partly a consequence of higher emphasis on research efforts in these areas, the North Atlantic continental shelves and slopes appear to represent particularly suitable grounds for the development of such reefs, but Lophelia occurrences have also been documented from the Gulf of Mexico in the western Atlantic (Schroeder, 2002; Ross & Nizinski, 2007; Brooke & Young, 2009; Hübscher et al., 2010; Larcom et al., 2014) and the US mid-Atlantic coast (Mienis et al., 2014; Brooke et al., 2017). The reported depths of L. pertusa occurrences encompass a broad bathymetric range from 39 m to over 3,300 m in the North Atlantic, but L. pertusa is most commonly found between 200 and 1,000 m (Roberts et al., 2009). L. pertusa inhabits temperatures between 4 to 13.9 °C and salinities from 32 to 38.8 (Freiwald et al., 2004, 2009).

Cold-water coral reefs are often characterised by colonies of different colour morphotypes (in the following short: colourmorphs). In the Norwegian L. pertusa reefs the typically white coral framework is accompanied by an orange phenotype of this species. Elde et al. (2012) found different pigment concentrations of astaxanthin, one of the main carotenoids in Lophelia, between the colourmorphs with more than twice the content in the soft tissue and skeleton of orange L. pertusa compared to the white phenotype. To date, it is unclear whether these variations in astaxanthin content are genetically or environmentally controlled in cold-water corals. Colour variation in organisms often results from different food sources (Elde et al., 2012). In the case of Lophelia it is rather unlikely that colour variation among colonies is related solely to food sources, since orange and white specimens grow side by side and have access to the same food. Instead, it may be related to differences in the composition of bacterial communities associated to the corals as reported for L. pertusa, which may be linked to a nutritional advantage of the corals (Neulinger et al., 2008). Pigmentation might also be an inherited characteristic transferred from individual to individual (Elde et al., 2012), as the orange pigment was also found to be vertically transmitted to the eggs of orange specimens (Larsson et al., 2014). Moreover, a role in the function as antioxidant or antibacterial agent was suggested as potential physiological advantage of these pigments to protect the corals against pathogens and to remove particles and sediment (Shnit-Orland & Kushmaro, 2008; Elde et al., 2012). In a recent study by Provan et al. (2016) the authors observed that the protein content of the mucus of L. pertusa varied between the two colour variants, which was suggested to be linked to the differences in mucus-associated bacterial symbionts.

Like many other deep-sea organisms, L. pertusa grows slowly, but has a long colony lifespan (Rogers, 1999). Many experimental laboratory studies report on growth rates of white L. pertusa under various conditions (Maier et al., 2009, 2012; Form & Riebesell, 2012; Lunden et al., 2014; Hennige et al., 2014a, 2015; Büscher, Form & Riebesell, 2017). Comparisons with growth rates in the natural habitat are scarce, as cold-water corals are difficult to access and in situ studies challenging to apply. Reported estimates of growth rates have a broad range from 2.4 to 35 mm per year, depending on regional differences and application of different methods, including potential sampling errors (for instance, in the complex skeletal sampling for isotope analyses) or lack of resolution (Duncan, 1877; Wilson, 1979; Mikkelsen et al., 1982; Freiwald, Heinrich & Pätzold, 1997; Mortensen, Rapp & Båmstedt, 1998; Bell & Smith, 1999; Roberts, 2002; Orejas, Gori & Gili, 2008; Orejas et al., 2011; Brooke & Young, 2009; Lartaud et al., 2013; Larcom et al., 2014).

In situ growth estimates of L. pertusa typically refer to linear and radial extension rates (reviewed in Rogers, 1999; Freiwald et al., 2004; Roberts et al., 2009). The majority of in situ growth rate estimates originate from non-invasive, indirect approaches through video and still inspection of corals grown on artificial substrates such as submarine cables and energy installations (Duncan, 1877; Wilson, 1979; Larcom et al., 2014), oil and gas platforms (Bell & Smith, 1999; Roberts, 2002; Gass & Roberts, 2006), and shipwrecks (Roberts et al., 2003; Larcom et al., 2014) that allow for estimates of minimum growth rates when considering the maximum age of coral colonies. Inspection of video surveys and still images revealed the highest reported growth estimates of 34–35 mm yr−1 (Gass & Roberts, 2006; Larcom et al., 2014), though these high growth rates of L. pertusa colonies from artificial substrates may be a result of very favourable conditions with regard to currents and food availability (Mortensen, 2001; Larcom et al., 2014).

Direct in situ measurements of linear extensions of L. pertusa were first performed by Brooke & Young (2009) by means of a 1-year mark and recapture approach in the northern Gulf of Mexico. The coral fragments from their experiment yielded far lower linear extension rates of 2.44–3.77 mm yr−1 on average compared to extensions from indirect analyses. Another mark and recapture study with L. pertusa from the Mediterranean Sea reported average linear extension rates of 7.5 mm yr−1 (Lartaud et al., 2013). No direct measurements of natural growth rates were previously reported from Norwegian cold-water coral reefs, although these reefs comprise the most densely populated cold-water reefs known (Freiwald et al., 2004). Laboratory investigations on calcification rates revealed more than an order of magnitude lower growth rates of L. pertusa from the Northeast Atlantic (0.006–0.009% d−1; Form & Riebesell, 2012; Büscher, Form & Riebesell, 2017) compared to the Mediterranean (0.02–0.03% d−1; Orejas et al., 2011; Maier et al., 2009, 2012).

Reef development is, however, not solely dependent on active growth of live corals. Cold-water coral reefs are characterised by a large proportion of dead coral framework that accounts for more than 70% of L. pertusa colonies (Vad et al., 2017). As for their warm-water relatives, also cold-water coral reef development comprises a balance of reef accretion and degradation. The proportion of living corals in an established colony was presumed to decrease compared to an increasingly higher amount of dead coral framework as a result of natural reef development (Vad et al., 2017). Therefore, it is important to consider the counter-acting processes of dissolution and particularly bioerosion of the dead coral framework when assessing reef growth and development.

Bioerosion is defined as ‘the process by which animals, plants and microbes sculpt or penetrate surfaces of hard substrates’ (Neumann, 1966; Bromley, 1994). In fossil and recent L. pertusa skeletons a variety of bioerosion traces such as borings, attachment scars, and grazing traces produced by a wide spectrum of organotrophic bioerosion agents (e.g. excavating sponges and bryozoans, microbial bioeroders including fungi and bacteria, grazing gastropods and echinoids, and parasitic foraminifers) was documented (Beuck & Freiwald, 2005; Bromley, 2005; Wisshak et al., 2005; Wisshak, 2008; Beuck, López-Correa & Freiwald, 2008; Beuck, Freiwald & Taviani, 2010). The majority of these bioeroders chemically etch and dissolve the host substrate. This process acts particularly on the bare dead coral skeleton, which lacks protection by organic tissue or defence mechanisms such as the secretion of mucus (Beuck, Freiwald & Taviani, 2010). Qualitative assessments of bioerosion patterns in L. pertusa have shown that for the majority of recorded bioerosion traces, that is, their trace makers, an exclusive or at least partial chemical mode of penetration is known or inferred. This also applies for the two most common organotrophic agents of bioerosion in these substrates, bioeroding marine fungi and excavating hadromerid sponges. For the latter group, which often take the lion share of internal macrobioerosion, several experiments with representatives of the most common genus, Cliona, have demonstrated a significant increase in bioerosion capacity with increasing carbon dioxide concentrations (Wisshak et al., 2012, 2013, 2014).

The growth of cold-water corals depends largely on environmental conditions including temperature, currents, food availability, and seawater chemistry (Mortensen, Rapp & Båmstedt, 1998; Dullo, Flögel & Rüggeberg, 2008; Flögel et al., 2014). Ongoing ocean change may affect the capability of these fragile organisms of building their calcareous skeletons, as ocean acidification caused by anthropogenic carbon dioxide (CO2) emissions results in lowered seawater pH and decreasing carbonate ion concentrations in the oceans and consequently in a diminished calcium carbonate (CaCO3) saturation state (Orr et al., 2005). Carbonate chemistry investigations at cold-water coral reefs and modelling assessments indicate that some L. pertusa habitats face already now low carbonate ion availability and projections suggest that ~70% of the known cold-water corals are expected to be exposed to calcium carbonate undersaturated waters by the end of the century due to ocean acidification (Guinotte et al., 2006; Zheng & Cao, 2015; Georgian et al., 2016). To make reliable predictions on the growth performance of cold-water corals and reef development in the future, it is therefore important to identify the natural range of growth rates and their thresholds at current ocean conditions and bring rates measured in situ in line with rates yielded from laboratory investigations. This may also help to assess if results yielded in the laboratory might represent over-targeted accretion as potential compensation response of the corals for suboptimal conditions, for example.

With regard to reef degrading processes under proposed future ocean conditions (IPCC, 2014), empirical studies on warm-water coral reef ecosystems suggest that bioerosion of CaCO3 will be accelerated in the future due to the promotion of chemical dissolution through lower coral skeletal densities under ocean acidification (Tribollet, Atkinson & Langdon, 2006; Tribollet et al., 2009; Wisshak et al., 2012, 2013, 2014; Reyes-Nivia et al., 2013; and see Schönberg et al., 2017 for a review). Based on such experimental data, model calculations have shown an alarming situation with regard to increasingly fragile carbonate balance of coral reefs and call for local and global action (Kennedy et al., 2013). These include conservation efforts and climate change mitigation strategies to prevent degradation of reef structures and eventually coral reef structural collapse. Despite the potentially significant effects, most studies regarding climate change related threats to corals do not consider bioerosion and almost all studies including degradation processes in coral reefs were carried out in tropical reef ecosystems. While studies have shown that bioerosion sometimes balances or even exceeds carbonate production temporarily in tropical reef ecosystems even at current ocean conditions (Perry, Spencer & Kench, 2008), to date there is no corresponding experimental data available for cold-water coral reefs. Studies of bioeroders from an intermediate, cold-temperate environment suggest that the observed patterns of accelerated bioerosion under future conditions may apply across species and latitudes (Wisshak et al., 2014). Hence, to gain a better understanding of baseline in situ bioerosion rates of cold-water coral reef substrates and to allow predictions with regard to the impacts of ocean change on reef degradation, it is crucial to include bioerosion analyses in studies assessing growth in cold-water coral habitats.

Thus, the aim of this study was to simultaneously assess in situ growth and bioerosion rates of L. pertusa in a 1 year mark and recapture experiment. We thereby compare two different methodological approaches, the change in weight over time (buoyant weight before and after deployment) and linear extension rates (determined via staining). In addition, we compare two different cold-water coral reef locations (off-shore vs. coastal reef) in mid-Norway, allowing for a better representation of the natural variability of environmentally differing reef settings. Moreover, live corals of white and orange L. pertusa were compared at each location in order to determine physiological differences between different colourmorphs in a reef with regard to their growth performance. Last but not least, normalisation of physiological rates will be expanded by determining volume and area of each coral fragment after recovery in addition to dry weight and biomass. This provides us with a variety of normalisation parameters and conversion factors between them, which can be used in future cold-water coral growth studies for more easily comparability.

Material and methods

Studied reef sites

For a 1 year in situ growth and bioerosion rate assessment, two Norwegian Lophelia reef sites with different environmental characteristics were chosen for collection and re-deployment of live corals and dead erect coral framework. The approximately 13 km long and 700 m wide Sula Reef Complex on the Sula Ridge off the coast of Sør-Trøndelag is the second largest known Lophelia reef on the Norwegian Shelf (Freiwald et al., 2002; Hovland et al., 2005). This offshore location comprises a relatively constant habitat in terms of environmental parameters such as temperature, salinity, pH, and currents, while the selected inshore location, a reef near the island Nord-Leksa in the outer Trondheimsfjord (henceforth referred to as Leksa Reef), is exposed to a highly variable environment due to strong tidal and compensatory currents (Form et al., 2015 Cruise Report POS473). At this location the in situ experiments were placed both in the living area of the reef and in the zone of dead coral debris a few tens of metres downslope.

Collection of Lophelia pertusa and maintenance on board

Sampling of coral specimens of the species L. pertusa was conducted with kind permission of the Norwegian Directorate of Fisheries (Fiskeridirektoratet) under permit number 12/17918. Corals from the Leksa Reef were collected on 29th and 30th June 2013 at 63°36.46′N and 9°22.76′E and 157 m water depth (white specimens) and 63°36.43′N and 9°22.45′E and 152 m (orange specimens) during research cruise POS455 with RV POSEIDON (GEOMAR Helmholtz-Zentrum für Ozeanforschung, 2015). At the Sula Reef, corals of both colourmorphs were collected on 4th July 2013 at 64°06.62′N, 8°07.1′E in 303 m water depth. At both sites, dead erect coral framework, bearing established bioeroder communities (chiefly bioeroding fungi, bacteria, bryozoans, and sponges), was sampled from the reef basis. All samples were collected by means of the manned submersible JAGO with its sensitive claw for non-destructive sampling (GEOMAR Helmholtz-Zentrum für Ozeanforschung, 2017). On board, corals and dead coral framework were placed in large holding tanks (120 × 110 × 80 cm) filled with 500 L natural seawater obtained from ~70 m depth. Four of those holding tanks were connected in order to create a recirculating system. An interconnected cooling unit (Titan 4000; Aqua Medic GmbH, Bissendorf, Germany) kept the water temperature in the tanks at ambient seabed temperature of 7.5–8.5 °C.

Preparation of the corals and dead erect coral framework for re-deployment

Live coral colonies as well as dead erect coral framework were fragmented into fist-sized pieces soon after collection. Afterwards, live corals were stained with Alizarin Red S. For this purpose, live coral branches were placed in separate staining tanks (2 × 30 L plastic containers) mounted within the holding tanks for temperature equilibration. The dye, pre-dissolved in ethanol, was slowly added until a concentration of 10–15 mg L−1 was reached according to the protocol applied in Brooke & Young (2009) for L. pertusa. Live specimens were incubated in the staining tanks for 2–3 days, as cold-water corals incorporate the dye more slowly than faster growing warm-water corals (Brooke & Young, 2009; Form & Riebesell, 2012). The dead framework was examined for calcifying epibionts, which were carefully removed with tweezers and a scalpel in order to eliminate weight gain due to their ongoing calcification during the experiment. All living and dead coral fragments were weighed under water following the buoyant weighing technique described by Davies (1989), employing a Sartorius BP 310 P (Göttingen, Germany; d = 0.001 g) with a purpose built free hanging weighing gondola to enable weighing on board the vessel and to reduce transmission of vibrations onto the balance. Weighing on board was performed at very calm sea, but was nevertheless unsteady. Therefore, the average of 10 consecutive values was taken for each fragment to improve precision. Shortly before deployment of the fragmented and stained corals, one white and one orange coral fragment were attached with cable ties inside a ‘coral cage’ (168 × 178 × 156 mm PP Nalgene® autoclave baskets). Four such coral cages were prepared for each study site.

Bioeroded coral fragments were attached in smaller baskets (123 × 154 × 105 mm, PP, Nalgene® autoclave baskets). For each location six ‘bioerosion cages’ filled with dead coral framework material were weighed under water using the buoyant weighing technique (Davies, 1989) and attached as a cluster for facilitating deployment and recovery. In bioeroded dead L. pertusa framework, a number of successive stages of bioerosion, characterised by certain bioerosion trace assemblages, have been identified (Beuck, Freiwald & Taviani, 2010). For the purpose of our experiment, we attempted to distribute skeletons of these different bioerosion stages (stages 3 to 6 sensu Beuck, Freiwald & Taviani, 2010) as evenly as possible to the different replicates and locations, though the amount of bioeroders per fragment can still vary considerably. As in Sula only little and relatively young dead coral framework could be collected, the material in the Sula cages comprised a mixture of dead erect coral framework from both collection sites in Sula and Nord-Leksa. Apart from this, we attempted to distribute skeletons as evenly as possible to the different replicates and locations with regard to their appearance of bioerosion stages.

Deployment and recovery of coral and bioerosion cages

Four coral cages and six bioerosion cages were deployed simultaneously at each of the three deployment locations, two at the inshore reef south of Nord-Leksa and one at the offshore Sula Reef in July 2013. To assure constant submersion in seawater, the cages were immersed in a sampling box installed in front of JAGO filled with seawater before lifting the submersible into water. On the ground, the baskets were positioned at the desired locations one by one with the submersible’s manipulator arm. At the first Leksa station (Leksa on-reef), the cluster and the coral cages were placed into a living reef area at 180 m water depth (Fig. 1A). At the second Leksa location (Leksa off-reef), baskets were placed on the bare sediment off the live reef zone at 218 m (Fig. 1B) to determine whether coral survival and growth is also supported in areas of no coral growth in the vicinity of the living reef. At the Sula Reef Complex (Sula), coral cages and the bioerosion cluster were deployed at the southernmost third of the reef chain in a small depression almost completely engulfed by the live reef at 304 m, about 50–100 m away from the nearest living Lophelia colonies. Unfortunately, in Sula no coral baskets could be deployed into the living reef structures because of limited dive possibilities due to rough weather conditions.

Figure 1 Live coral and bioerosion cages deployed within living reef structures (on-reef, (A)) and on the sediment in the off-reef location (B) in Nord-Leksa.

Image courtesy: JAGO-Team, GEOMAR Kiel.

After 1 year, in July/August 2014, all locations were revisited with RV POSEIDON (POS473) and all coral cages and bioerosion clusters were recollected by means of JAGO and brought back aboard (permit number 14/1781 of Directorate of Fisheries for cruise POS473), where they were immediately transferred to large holding tanks. Except for one coral cage in the Leksa off-reef location, all coral cages and clusters could be retrieved. Soon after recovery, the coral fragments and bioerosion cages were weighed on board following the same protocol and using the same equipment as outlined above. Afterwards, all samples were dried at 60 °C on board and packed cushioned for later laboratory analyses.

Post-cruise analyses

After the cruise, all samples were dried at 70 °C for several days until constant weight was reached. Dry weights of empty baskets, cable ties, and corals were separately measured. Calcifying epibionts grown on the dead erect coral framework as well as on the live coral fragments (carbonate accretion) over the year of exposure were removed with a scalpel and weighed separately after drying to constant weight. Then, all samples were scanned via computed tomography (CT) for volume and surface area analysis (see detailed description of the methods below), before linear extension rates of the ‘live’ coral fragments were determined by measuring the distance from the Alizarin Red S band of each corallite to the rim of the calyx using a digital calliper. As growth of the calices is sometimes more pronounced on one side of the calyx than on the other, this measurement was done on two opposing sides, with the lowest and the highest distance between stain bands and rim of each corallite. In addition, the numbers of newly grown (completely unstained) corallites, as well as the number of died polyps (calices without tissue) of the ‘live’ coral fragments were counted. When being deployed in 2013 it was made sure that only intact corallites remained on the fragments, while all empty corallites were removed. Counted newly grown corallites and dead polyps were compared to the total number of corallites of each branch to assess budding rate as well as mortality as a percentage over the course of the experiment. Finally, all samples were dried again to constant weight and combusted at 500 °C for 5 h for differentiation of organic vs. inorganic content.

All fragments were dried to constant weight at 68 °C before tissue residuals were removed with chlorine bleach according to the method described by Davies (1989). Afterwards, the buoyant weight of the fragments without tissue residuals was measured. In order to get rid of all accumulated air bubbles within the skeletal structures, fragments were treated in a vacuum drying cabinet in beakers filled with seawater, so that the weights were not falsified by additional buoyancy. After being washed in distilled water, the fragments were dried and weighed again until constant weight was reached and the skeletal densities were calculated from the following equation following the method by Davies (1989): δSkeleton=δsw(1−BWDW),

with δsw = density of the seawater, BW = buoyant weight of the coral fragments without tissue, and DW = the dry weight of the fragments without tissue.

CT scanning

Computed tomography scans of all dried samples (‘live’ coral fragments and dead framework) for volume and surface area analyses were carried out with a Toshiba Aquilion 64 computer tomograph at the hospital Klinikum Bremen-Mitte with a voltage of the X-ray source of 120 kV and a current of 600 mA. The resulting CT image stacks have a resolution of 0.35 mm in x- and y-direction and 0.5 mm resolution in z-direction (0.3 mm reconstruction unit). Images were reconstructed using Toshiba’s patented helical cone beam reconstruction technique and are provided in DICOM-format. The data were processed with the ZIB edition of the Amira software (version 2013.47; ZIB, Berlin, Germany) (Stalling, Westerhoff & Hege, 2005). With Amira, the corals were segmented with the Multi-Thresholding module (threshold value: 0). The segmentation result was evaluated and the coral cage was removed from the computation of each sample with the Segmentation Editor. Afterwards, the Generate-Surface module was used to compute a surface model of the coral specimens. Finally, the volume and the area of the specimens were determined using the Surface Area Volume module.

Calculations and statistical analyses

Growth and bioerosion rates were calculated according to descriptions in Davies (1989) based on buoyant-weight gain or loss of the coral skeleton over time. The gained rates were normalised to weight change per day as a percentage of the initial weight of the coral (G % d−1) as parameterisation predominantly used in experimental studies with live corals applying the buoyant weighing technique. In addition, rates were normalised to weight change in grams per square metre coral surface (gained from the CT measurements) per year (g m2 yr−1), which represents the most common unit in bioerosion studies. Data are depicted as mean ± standard deviation (SD). Statistical analyses were performed using SigmaPlot© (version 12.0; Systat Software, Inc., San Jose, CA, USA) and MS Excel Redmond, WA, USA. For statistically comparing the results between the three locations of white as well as orange coral colourmorphs, One-way analysis of variance tests (ANOVAs) were carried out with n = 4 replicates per location, except for the Leksa off-reef location at which one basket and therewith one orange and one white replicate were missing. In case of statistical differences, a post-hoc test for pairwise multiple comparisons following the Holm-Sidak method was carried out to distinguish differences among groups/locations. Whenever data were pooled to increase the sample size and statistical power, this was done upon confirmation that there were no significant differences in the statistical tests among pooled groups. For direct comparisons of white and orange corals or only two locations, t-tests were performed. In order to obtain more accurate and reliable means for conversion factor calculations, outlier tests were carried out in MS Excel (Excel QUARTILE and OR functions).

Results

Coral structural analyses

Coral surface area, volume, corallite number, and skeletal density of live and dead coral fragments (Table 1) were gathered post-experiment after recovery of the coral and bioerosion baskets. Volume and surface area were significantly different between live corals from both Leksa sites and the Sula Reef in both colourmorphs (p ≤ 0.001; One-way ANOVAs), with significantly less bulky coral fragments deployed in Sula compared to both Leksa locations. While Leksa corals had 119 ± 44 polyps/corallites per fragment on average, the Sula corals had only 35 ± 11 polyps per branch (Table 1). This is attributable to the different morphology of the offshore corals. While fjord colony growth is more compact, offshore coral growth tends to be more extended and branched, which corresponds to lower polyp numbers as well as surface area and volume in Sula despite similar fragment sizes like the Leksa fragments. Both surface area and volume of the coral branches correlated well with polyp count (R2 = 0.7) with a slightly better correlation of surface area with total polyp count than volume. Figure 2 shows exemplary CT scan images of a live coral fragment (A), and dead coral framework (B) from one basket of the cluster.

Figure 2 Example images of the CT scans of (A) a live coral fragment (orange coral branch from Leksa on-reef) and (B) dead coral framework from one basket of the cluster (Leksa off-reef).

Table 1 Coral surface area, volume, and skeletal density of live and dead coral fragments as well as polyp counts of live corals.

Surface area (in mm2) and volume (in mm3) are calculated from CT scans and given for live white and orange coral fragments as average of all replicates (n = 4 white as well as orange corals at Leksa on-reef and Sula, and n = 3 white as well as orange corals at Leksa off-reef) ± standard deviation at the three deployment locations.

Location	Replicates	Surface area (mm2)	Volume (mm3)	Polyps/corallites (N)	Skeletal density (g cm−3)	
Leksa (inshore, on-reef)	White
corals	51,074.1 ± 1,793.9	36,403.1 ± 5,607.8	126 ± 55	2.764 ± 0.011	
Orange
corals	30,064.7 ± 11,692.6	21,150.5 ± 11,044.0	91 ± 10	2.733 ± 0.057	
Dead
framework	1,72,819.7 ± 12,298.6	1,44,023.3 ± 13,345.2	–	2.777 ± 0.024	
Leksa (inshore, off-reef)	White
corals	44,529.3 ± 8,823.8	34,855.6 ± 7,881.5	111 ± 33	2.746 ± 0.031	
Orange
corals	48,469.5 ± 23,321.8	32,660.8 ± 17,937.6	154 ± 57	2.700 ± 0.022	
Dead
framework	1,69,054.3 ± 11,319.0	1,41,679.3 ± 13,145.4	–	2.770 ± 0.010	
Sula (offshore, off-reef)	White
corals	20,323.5 ± 7,228.6	13,304.7 ± 5,139.5	38 ± 10	2.722 ± 0.062	
Orange
corals	16,063.2 ± 5,236.2	11,052.1 ± 3,955.8	33 ± 12	2.411 ± 0.135	
Dead
framework	1,41,404.2 ± 15,298.4	1,10,715.6 ± 11,055.5	–	2.727 ± 0.030	

Mean skeletal density of all live corals was 2.734 ± 0.043 g cm−3. Orange corals had a slightly lower skeletal density (<1%) by trend than white corals. Note that the skeletal density of the orange coral fragments from the Sula Reef was considerably lower than the densities of all other fragments and was identified as outliers. The outlier values were therefore omitted from the average skeletal density of live corals. Bioeroded skeleton material had slightly higher densities than live corals averaging 2.758 ± 0.031 g cm-3. Both, white vs. orange live as well as bioeroded vs. live coral skeletons were not significantly different in densities (t-tests). For the calculation of growth and bioerosion rates the specific density means of live or bioeroded skeleton material was used.

Mortality of live Lophelia fragments

Polyp mortality of the branches was quite variable between the replicates within and among locations, ranging from 0–86% dead polyps per branch. The highest variability was found in the Leksa on-reef location (Fig. 3). There was no statistically significant difference in mortality between locations (white and orange live coral branches separately tested or pooled). However, lowest mortality was found in Sula with only half as many dead polyps as a percentage of the total polyp count of a branch (10 ± 14%) as in the Leksa off-reef location (21 ± 19%) and one third of the percentage of the Leksa on-reef group (30 ± 27%). Comparison of polyp mortality between white and orange fragments (Fig. 4) revealed a statistically significant difference when white and orange corals were pooled over all three locations (p = 0.002; Mann–Whitney Rank Sum Test). While the orange coral fragments had on average 8 ± 9% dead polyps per replicate, the white corals had 33 ± 23% (Table 2).

Figure 3 Mortality (in percent dead polyps per branch) of white and orange corals at three deployment sites.

Error bars represent ± standard deviation (SD) of white and orange corals each at the inshore on-reef and off-reef deployment locations at Nord-Leksa and at the Sula Reef.

Figure 4 Mortality (in percent dead polyps per branch) of white and orange corals averaged over all three locations.

Error bars represent ± SD. The asterisk denotes that there is a statistically significant difference (p = 0.002) of the percentage of polyp mortality between white and orange corals.

Table 2 Percent mortality of white and orange coral branches after 1 year of deployment.

The percentage of polyp mortality per branch is given as replicate means ± standard deviation for each deployment location and colourmorph.

Location	Mortality rate
white (%)	Mortality rate
orange (%)	
Leksa (inshore, on-reef)	48 ± 26	12 ± 14	
Leksa (inshore, off-reef)	33 ± 19	9 ± 7	
Sula (offshore, off-reef)	18 ± 17	3 ± 5	
Mean of all locations	33 ± 23	8 ± 9	
Mean all live corals	20 ± 22	

Linear extension rates

The overall mean extension rate of all stained corallites of the living coral branches (not all corallites incorporated the dye) from all sites was 2.12 ± 0.86 mm yr−1 (n = 18; Table 3). Examples of coral branches with corallites showing the Alizarin Red S band are depicted in the photographs in Fig. 5. There were no statistically significant differences in average linear extension rates of the replicates, neither between the three locations (One-Way ANOVA) nor between colourmorphs (t-test) (Fig. 6). Nevertheless, the orange specimens tended to have ~15% higher linear extensions than the white ones (pooled over all locations: 2.31 ± 0.90 mm yr−1; n = 8 (orange) vs. 1.96 ± 0.84 mm yr−1; n = 10 (white); Table 3). However, especially the orange corals showed high variances between replicates, which was pronounced most strongly in the on-reef replicates of Leksa, similarly to weight gain. Moreover, Sula corals showed considerably lower growth (~46% less average linear extension) compared to inshore sites. Within the orange coral group this is, however, based on only one replicate of the Sula location as the dye was visibly incorporated in only one of the four replicates at Sula. Thus, averaging all Leksa corals regardless of white or orange from both Leksa sites and comparing Leksa and Sula extension rates revealed a statistically significant difference with ~44% higher extension rates (p = 0.03; t-test; 2.34 (n = 14) vs. 1.31 mm yr−1 (n = 4); Fig. 7). Average linear extension rates correlated well with weight gain in percent per day (R2 = 0.83).

Figure 5 Exemplary branches of the stained live corals of a white (A + B) and an orange (C + D) coral from the Nord-Leksa on-reef location.

Photos (B) and (D) depict close-ups of the most distant polyps of the coral branches from (A) and (C). Linear extension rates were measured from the Alizarin Red S staining mark to the terminal, unstained rim of the calices.

Figure 6 Average linear extension rates (in mm per year) of white and orange corals over 1 year at the three deployment sites Leksa on-reef, Leksa off-reef, and Sula Reef.

Number of replicates (n) per site and colour morphotype is indicated next to the bars. Error bars represent ± standard deviation.

Figure 7 Average linear extension rates (in mm per year) of inshore (Leksa) and offshore (Sula) live corals.

Given are mean extension rates of pooled white and orange corals in mm per year ± standard deviation as error bars. The asterisk denotes a statistically significant difference between inshore and offshore corals when white and orange corals are pooled at both locations (p = 0.03).

Table 3 Calcification and linear extension rates of live corals at three different deployment sites over 1 year experimental duration.

Rates (in percent per day (% d−1) as well as grams per square metre and year (g m−2 yr−1) for weight gain and in mm per year (mm yr−1) for linear extension) are given as replicate means ± standard deviation per deployment site for white and orange corals. The calcification rate in g m−2 yr−1 is based on surface area of the coral substrate calculated from the CT scans (see text for details).

Location	Calcification rate—white
(% d−1)	Calcification rate—orange
(% d−1)	Calcification
rate—white
(g m−2 yr−1)	Calcification
rate—orange
(g m−2 yr−1)	Linear
extension—white
(mm yr−1)	Linear
extension—orange
(mm yr−1)	
Leksa (inshore, on-reef)	0.0109 ± 0.0086	0.0202 ± 0.0203	56.07 ± 42.31	99.31 ± 86.73	1.98 ± 0.57	2.44 ± 1.26	
Leksa (inshore, off-reef)	0.0120 ± 0.0058	0.0165 ± 0.0085	74.01 ± 38.94	76.80 ± 45.83	2.65 ± 1.06	2.38 ± 0.33	
Sula (offshore, off-reef)	0.0095 ± 0.0030	0.0053 ± 0.0030	42.49 ± 6.82	28.51 ± 15.03	1.25 ± 0.25	1.54 (n = 1)	
Mean of all locations	0.0107 ± 0.0057	0.0138 ± 0.0137	56.02 ± 32.01	67.42 ± 61.51	1.96 ± 0.84	2.31 ± 0.90	
Mean all live corals	0.0122 ± 0.0103		61.72 ± 48.20		2.12 ± 0.86		

The amount of newly grown corallites that developed after staining was similar in all locations and averaged 47.4 ± 12.5%. New corallites alone had significantly higher extension rates (p = 0.043; t-test) than all stained polyps of the branches (total extensions; including newly grown polyps and all corallites where staining bands could be determined). Mean linear extension of newly grown polyps/corallites over all locations and specimens was 2.84 ± 1.04 mm yr−1 compared to 2.11 ± 0.86 mm yr−1 total extension on average (Fig. 8), and compared to 1.87 ± 0.59 mm yr−1 when taking only the ‘old’ stained corallites alone (p = 0.003; t-test comparing newly grown and stained corallites excluding newly grown). Comparing new vs. total extensions (averaged over all stained and newly grown corallites) of the different groups shows that the greatest effect of new growth took place in the Leksa on-reef location. On-reef, 60–75% higher linear extension rates of new corallites of white and orange specimens were gained, while in the Leksa off-reef location it was less than half as much (20–33%). In Sula, growth rates of newly grown corallites were not different or even lower than total stained corallites, although the percentage of newly grown corallites per branch was similar to the percentage of newly grown polyps on the Leksa branches. As newly grown corallites make up for almost half of all stained corallites the pattern of the slight differences between location and/or colourmorph is similar to average extensions of all (old and young) corallites.

Figure 8 Average linear extension rates (in mm per year) of total stained polyps and polyps grown newly after staining.

Data are averaged over all three reef sites. Error bars represent ± standard deviation. The asterisk denotes that there is a statistically significant difference between only newly grown polyps and all stained polyps in average linear extension (p = 0.04).

Calcification rates of live corals

Overall net carbonate production rate of all observed live coral fragments based on buoyant weight measurements (SD of the 10 consecutive measurements of each fragment of the Leksa weighing session = 0.052 g and Sula weighing session = 0.082 g) was 0.0122 ± 0.0103% d−1 or 61.7 ± 48.2 g m−2 yr−1 (n = 22; Fig. 9; Table 3). Mean values of white and orange coral fragments averaged over all locations were 0.0107 ± 0.0057% d−1 or 56.02 ± 32.01 g m−2 yr−1 (n = 11) and 0.0138 ± 0.0137% d−1 corresponding to 67.42 ± 61.51 g m−2 yr−1 (n = 11), respectively. Calcification rates did not differ statistically significantly between white and orange corals (averaged over all locations; t-test), nor between the different sites (One-way ANOVAs). However, corals from Sula Reef generally showed lower calcification rates with only half as much CaCO3 precipitation on average (48%) as the Leksa off-reef corals when comparing all live corals (white and orange pooled) of the single sites (p = 0.029; t-test). Although growth rates in the top reef replicates in Leksa on-reef were even higher on average than at the Leksa off-reef site, the top reef corals were not significantly different from the Sula corals. A similar comparison of all inshore vs. offshore replicates as for the linear extension rates (compare Fig. 7) revealed a similar picture with 50% higher growth rates in weight gain in % d−1, though not statistically significant due to the high variability in the Leksa on-reef location (0.015% d−1 (n = 14) vs. 0.0074% d−1(n = 8)). Differences between the different locations and between the two colourmorphs of L. pertusa are shown in Fig. 10.

Figure 9 Growth rates (calcification) of live corals and bioerosion and accretion rates of dead coral framework averaged over all three deployment sites.

Given are mean net calcification rates of live corals averaged over all locations and colourmorphs as well as bioerosion rates of the dead erect coral fragments and accretion rates of calcifying fauna on the dead framework averaged over the deployment sites. Error bars represent ± standard deviation and the number of replicates (n) is indicated next to the bars.

Figure 10 Growth (calcification) and bioerosion rates of 1 year in situ investigation at the three deployment sites Leksa on-reef, Leksa off-reef, and Sula Reef.

Average calcification rates of white and orange L. pertusa, and bioerosion rates of dead coral framework and associated carbonate accretion by calcifying epibionts (in percent per day) over 1 year of exposure. Error bars represent ± standard deviation and the number of replicates (n) is indicated next to the bars, respectively.

Bioerosion rates and epibiont carbonate accretion rates of dead erect coral framework

Carbonate degradation rates presumably resulted primarily from bioerosion processes, as the aragonite saturation of the seawater was supersaturated at all locations at the time of deployment as well as recovery of the cages (Ω > 1.7; Table 4) and seasonal undersaturation (Ω < 1) is expected very unlikely. Thus, physicochemical dissolution of the corals’ skeleton is considered negligible here and degradation rates are referred to as bioerosion rates in the following. Bioerosion rates of the dead erect framework integrated over all locations (expressed here as negative values for indicating a loss in weight as opposed to the gain in weight by coral calcification) was −0.0020 ± 0.0015% d−1, corresponding to −12.37 ± 9.40 g m−2 coral surface yr−1 (Fig. 9). Highest degree of bioerosion took place in the Leksa on-reef location with −0.0036 ± 0.0012% d−1 or −23.20 ± 7.87 g m−2 yr−1, which was 74% higher than the off-reef site in Leksa with −0.0009 ± 0.0007% d−1 or −5.88 ± 4.42 g m−2 yr−1 and 64% higher than the offshore location Sula with −0.0013 ± 0.0003% d−1 or −8.03 ± 2.24 g m−2 yr−1. Values from the on-reef Leksa location were statistically different from both off-reef placements (p ≤ 0.001; One-Way ANOVA; Fig. 10; Table 5), despite highest variation between replicates in the on-reef site. The off-reef locations (in Leksa and Sula) were not significantly different from one another.

Table 4 Carbonate chemistry and physical seawater properties at three deployment locations (inshore at two sites in the Trondheimsfjord (Leksa on- and off-reef) and offshore at the Sula Reef).

Environmental seawater properties measured from samples taken directly at the coral cages deployment sites at the time of recovery in 2014. Water samples were collected by means of the NISKIN bottle of JAGO for measurements of total alkalinity (TA) and dissolved inorganic carbon (DIC) (in μmol per kg seawater). Physical seawater parameters (temperature (T) and salinity (Sal)) were measured with a CTD attached to JAGO (GEOMAR Helmholtz-Zentrum für Ozeanforschung, 2017). Remaining carbonate chemistry parameters (pCO2, bicarbonate (HCO3−), carbonate (CO32−), and the aragonite saturation (ΩAr)) were computed with CO2SYS.

Location	Latitude	Longitude	Depth (m)	T
(°C)	Sal	DIC
(μmol kg−1)	TA
(μmol kg−1)	pHTS	pCO2
(μatm)	HCO3− (μmol kg−1)	CO32− (μmol kg−1)	ΩAr	
Nord-Leksa ‘on-reef’	63°36.486′N	09°22.947′E	180	7.7	35.1	2,157.0	2,306.7	7.994	453.5	2,022	113.6	1.7	
Nord-Leksa ‘off-reef’	63°36.535′N	09°22.891′E	219	7.7	35.4	2,136.3	2,310.0	8.052	390.7	1,990	127.8	1.9	
Sula Reef	64°06.643′N	08°07.060′E	304	7.5	35.5	2,142.6	2,312.9	8.043	399.8	1,998	125.8	1.9	

Table 5 Bioerosion and carbonate accretion rates of the dead coral framework over 1 year experimental duration at one offshore and two inshore deployment sites.

Rates are given in percent bioerosion as well as carbonate accretion per day (% d−1), and in grams per coral framework surface and year (g m−2 yr−1). Data are presented as replicate means ± standard deviation for all three locations (Leksa on-reef, Leksa off-reef, Sula Reef).

Location	Bioerosion rate
(% d−1)	Accretion rate
(% d−1)	Bioerosion rate
(g m−2 yr−1)	Accretion rate
(g m−2 yr−1)	
Leksa (inshore, on-reef)	−0.0036 ± 0.0012	0.0042 ± 0.0016	−23.20 ± 7.87	26.58 ± 9.74	
Leksa (inshore, off-reef)	−0.0009 ± 0.0007	0.0027 ± 0.0003	−5.88 ± 4.42	17.13 ± 2.45	
Sula (offshore, off-reef)	−0.0013 ± 0.0003	0.0019 ± 0.0006	−8.03 ± 2.24	11.74 ± 3.41	
Mean of all locations:	−0.0020 ± 0.0015	0.0029 ± 0.0013	−12.37 ± 9.40	18.48 ± 8.54	

Carbonate accretion by calcifying epibionts that grew during the 1 year of exposure was 0.0029 ± 0.0013% d−1 (18.48 ± 8.54 g m−2 yr−1), accounting for about one-fourth (23.7%) of the growth in percent per day of the living corals (Fig. 9; Table 5). However, this number has to be taken with caution, as particularly accretion might be subject to estimation errors. Intensity of carbonate accretion in the dead framework was found to covary with the observed bioerosion rates, with highest accretion in the Leksa on-reef location and lower accretion rates in both off-reef sites. Carbonate accretion differed significantly only between the Leksa on-reef location and the Sula Reef (p = 0.006; One-Way ANOVA; Table 5; Fig. 10).

Similar to the growth rates of living corals, bioerosion as well as accretion rates showed the highest variability of rates (highest SD of the mean) at the on-reef location.

Conversion factors

We used various methods for growth rate measurements as well as for the normalisation of the different variables, and are thereby able to provide conversion factors for coral growth in size and weight and for the standardisation of these data (Table 6). Since there were no statistically significant differences between colourmorphs and locations, conversion factors for growth rates based on differences in buoyant weight or linear extension rates as well as buoyant weight vs. dry weight, dry weight vs. volume and surface area, and weight, volume or surface area vs. number of polyps were averaged across all samples of live corals. Weight, size, and polyp number correlated well (R2 ranging from 0.616 to 0.999).

Table 6 Calculated conversion factors of growth rates and structural parameters.

Factors translating growth rates of living corals from daily percentage (% d−1/g m−2 yr−1) into linear extension (mm d−1/yr−1) and vice versa, as well as conversions of different standardisation parameters such as weight, volume or surface area (short ‘Area’), and polyp count (N of polyps). Conversions are given for white and red corals separately and for both combined (‘All live corals’).

Parameters	White corals	Orange corals	All live corals	
G (% d−1) → Linear extension (mm d−1)	0.439 ± 0.125	0.456 ± 0.194	0.447 ± 0.155	
Linear extension (mm d−1) → G (% d−1)	2.040 ± 0.991	2.325 ± 1.060	2.167 ± 1.056	
G (g m−2 yr−1) → Linear extension (mm yr−1)	0.035 ± 0.019	0.038 ± 0.015	0.037 ± 0.017	
Linear extension (mm yr−1) → G (g m−2 yr−1)	30.34 ± 15.30	31.28 ± 14.03	30.76 ± 14.43	
Area (mm2) → Volume (mm3)	0.709 ± 0.095	0.677 ± 0.070	0.693 ± 0.083	
Volume (mm3) → Area (mm2)	1.435 ± 0.201	1.492 ± 0.153	1.464 ± 0.177	
Volume (mm3) → Dry weight (g)	0.002 ± 0.000	0.002 ± 0.000	0.002 ± 0.000	
Dry weight (g) → Volume (mm3)	501.88 ± 23.29	515.27 ± 23.80	508.58 ± 23.98	
Area (mm2) → Dry weight (g)	1,422.24 ± 248.19	1,321.02 ± 194.56	1,371.63 ± 223.70	
Dry weight (g) → Area (mm2)	724.16 ± 134.19	772.03 ± 113.52	748.09 ± 123.58	
N of polyps → Dry weight (g)	0.672 ± 0.210	0.522 ± 0.219	0.597 ± 0.223	
DW (g) → N of polyps	1.566 ± 0.517	1.998 ± 0.731	1.772 ± 0.650	
N of polyps → Area (mm2)	474.03 ± 128.58	385.58 ± 124.65	429.80 ± 131.61	
Area (mm2) → N of polyps	0.002 ± 0.001	0.003 ± 0.001	0.003 ± 0.001	
N of polyps → Volume (mm3)	334.91 ± 97.04	265.49 ± 104.09	300.20 ± 104.43	
Volume (mm3) → N of polyps	0.003 ± 0.001	0.004 ± 0.002	0.004 ± 0.001	
Buoyant (g) → Dry weight (g)	1.536 ± 0.008	1.536 ± 0.005	1.536 ± 0.011	
Dry weight (g) → Buoyant weight (g)	0.651 ± 0.004	0.651 ± 0.002	0.651 ± 0.005	

Discussion

In this in situ study, net growth- as well as bioerosion rates from environmentally contrasting cold-water coral ecosystems were obtained in a 1 year experiment in a Norwegian fjord and open shelf coral reef environment in the Northeast Atlantic using complementing established standard methods. In situ net calcification rates of healthy calcifying living L. pertusa of different morphological structure and colour were 0.011% d−1 on average over three different reef sites. Bioerosion rates of dead erect coral framework averaged −0.002% d−1 (−12.37 g m−2 yr−1) of the same reef habitats (disregarding the relatively high accretion rates of epibionts). Calculating a net production rate of accretion and bioerosion of live and dead coral fragments of this experiment is not legitimate, as here two different sample sizes are compared. For this reason we refrained from adding up calcification rates of living corals and erosion rates of dead coral framework, since the obtained rates do not reflect the actual reef carbonate budget, which would demand the determination of the proportions of live and exposed dead coral framework in a given reef. However, considering that bioerosion rates amounted to one fifth to one sixth of calcium carbonate loss compared with the production rates through coral calcification in this direct comparison, it can be assumed that the carbonate balance of accretion and bioerosion in cold-water coral reefs could be shifted towards higher biogenic dissolution in the future under ocean change. Supposing that the relative proportions in this one-to-one relationship are roughly correct and reef base degradation increases with acidification (compare Tribollet, Atkinson & Langdon, 2006; Tribollet et al., 2009; Wisshak et al., 2012, 2013, 2014; Reyes-Nivia et al., 2013) this might even lead to net negative carbonate budget states. Some tropical coral reefs experience already now negative net balanced carbonate budgets (Perry et al., 2014). While in some areas cold-water corals are found to grow in unfavourable conditions with regard to carbonate chemistry (Gómez et al., 2018), it is not known for these systems whether they are in net production or dissolution state. With regard to projected changes in seawater conditions in the future, bioerosion is expected to increase in warm-water reef systems, particularly driven by sponges (Wisshak et al., 2012, 2013). Considering a doubling to quadrupling of chemical bioerosion through sponges in warm-water reefs under different end-of-the-century pCO2 scenarios (Wisshak et al., 2013), cold-water coral reef resilience could become significantly impacted by bioerosion under ongoing ocean acidification, if similar enhanced bioerosion effects will become apparent in cold-water coral framework. Indeed, enhanced degradation may particularly become apparent in cold-water coral reefs considering that the calcium carbonate saturation in cold, deep waters is considerably lower than in shallow tropical environments. Since likely higher bioerosion rates can be expected from older Lophelia framework stages in the centre of the reef structures than from the material used here, a quadrupling of bioerosion rates could lead to higher degradation than accretion gained through calcification in cold-water coral reefs in a high CO2 world.

Growth and mortality of living corals

Growth rates

Net growth rates of live corals in the experiment ranged from 0.001 to 0.049% d−1 in weight gain (calcification) and 1.00–4.05 mm yr−1 in average linear extension. These values are in the lower range of literature values both in terms of buoyant weight measurements and length growth assessments (Brooke & Young, 2009; Maier et al., 2009; Orejas et al., 2011; Lunden et al., 2014; Büscher, Form & Riebesell, 2017). Buoyant weight measurements were usually applied in laboratory experiments, since on-board weighing comprises a difficult task due to the ship’s vibration and movement. Moreover, weighing of slow-growing coral fragments in particular requires very precise underwater weighing and a very careful handling during deployment and recovery of the weighed samples to not lose any bits of the corals by breakage. With a mean value of 0.011% d−1 for white coral specimens measured in this in situ experiment, growth rates are slightly higher than rates obtained in the lab with corals from different Norwegian reef sites ranging from 0.006% d−1 for corals from Nord-Leksa (Büscher, Form & Riebesell, 2017), over 0.007% d−1 for Oslofjord corals to 0.009% d−1 for Sula Reef corals (Form & Riebesell, 2012) at ambient temperatures. Higher growth rates of Norwegian corals were obtained by specimens kept at an elevated temperature of 12 °C (0.006–0.029% d−1, Büscher, Form & Riebesell, 2017), which were in a comparable range with rates of L. pertusa from the Mediterranean Sea at similar temperature (0.02 ± 0.01% d−1; Orejas et al., 2011; Maier et al., 2009, 2012). Those relatively high growth rates were also reached by some individual Norwegian specimens in the present in situ experiment at 7–8 °C. Short-term (14 days) calcification rates of L. pertusa from the Gulf of Mexico measured in the laboratory with the buoyant weighing technique were more than twice as high (0.025 ± 0.006% d−1, n = 16) on average (Lunden et al., 2014), while a longer-term study (6 months) with corals from this area yielded rates at ambient conditions that were comparable to Norwegian corals (Kurman et al., 2017). Lunden et al. (2014) observed quite a broad range of average calcification rates in their experiment, ranging from 0.002 to 0.091% d−1. Such rather high variances between fragments even at similar environmental conditions as likewise observed in this study display the very high plasticity of L. pertusa with regard to its performance, which was experienced in several investigations of this species (Mortensen, 2001; Brooke & Young, 2009; Form & Riebesell, 2012; Lunden et al., 2014; Hennige et al., 2015; Büscher, Form & Riebesell, 2017; Kurman et al., 2017). Kurman et al. (2017) found varying growth responses of different genotypes of L. pertusa exposed to acidified conditions, with some genotypes withstanding the same conditions longer than others. The authors hypothesised that some genotypes may prove to be more resilient towards ocean change than others showing that L. pertusa may contain the genetic variability necessary to support adaptive responses to changing conditions in the future (Kurman et al., 2017). Thus, the broad range in growth rates may also result from genetic variability across colonies.

With regard to linear extensions, it is rather difficult to compare measured rates with literature values obtained from different methods (video survey, isotopic fractionation, artificial substrate observations, mark and recapture) and with different perspectives or intentions (i.e. determining the highest possible growth rate vs. average growth rate). Growth estimates from indirect analyses vary widely with extension rates from approximately 3.2–34.7 mm yr−1 (Mikkelsen et al., 1982; Freiwald, Heinrich & Pätzold, 1997; Mortensen, Rapp & Båmstedt, 1998; Bell & Smith, 1999; Roberts, 2002; Gass & Roberts, 2006; Larcom et al., 2014). The broad range of extension rates indicates site-specific differences, although it cannot be excluded that the different sampling techniques might contribute to the variations in results. While non-destructive indirect methods like video surveys might lead to both, underestimation of growth rates with unknown initiation of colony development as well as overestimation due to determination of only the largest and fastest growing colonies, direct analyses through, for example, mark and recapture approaches likely yield underestimated growth rates because of the handling impacts to the corals. For example, the staining of the corals with Alizarin Red S was argued to impact the coral’s growth recovery, leading to underestimated in situ natural growth of cold-water corals even in long-term approaches (Lartaud et al., 2017).

Highest linear extension rates were obtained through visual inspections via video surveys from corals grown on artificial substrates with known time of installation with maximum reported rates of 34.7 mm yr−1 in an individual colony (Gass & Roberts, 2006; Larcom et al., 2014). While the average of the largest corals from various platforms and depths (300–800 m) of the same study by Larcom et al. (2014) comprise already lower rates of 21 mm yr−1, this reduces even further to about 17 mm yr−1 when considering the largest 10% of all colonies (Larcom et al., 2014). Although the estimated growth rates might comprise an underestimation of potential maximum growth, since the calculations assume immediate settling after installation of the structures, the average extension rates in the mentioned studies likely represent above-average growth rates of the largest and most elongated colonies. On the other hand, corals growing on artificial substrates are situated in an exposed position well above the seafloor with unimpeded access to food particles and directed into the currents (Mortensen, 2001; Gass & Roberts, 2006; Larcom et al., 2014), which might support exceptionally high extension rates in individual colonies. Thus, coral growth on man-made structures constitutes in a way pioneer growth, with colonies being able to extend in all directions and therefore gaining very high extension rates, whereas in already established reefs it was observed that growth and the proportion of living L. pertusa decrease with age of the colony (Mortensen, 2001; Brooke & Young, 2009; Lartaud et al., 2013; Larcom et al., 2014; Vad et al., 2017).

Linear extension rates measured in laboratory studies on L. pertusa were usually lower than the field observations. Highest extension rates of 15–17 mm yr−1 were obtained from aquarium cultivations of L. pertusa from the Mediterranean Sea (Orejas, Gori & Gili, 2008). In an elongated observation of this study, however, Orejas et al. (2011) measured a mean linear extension rate of ∼ 9 mm yr−1, which is in accordance with linear extensions measured of L. pertusa from different regions and through different analytical methods ranging from ~5.5 to 9.5 mm yr−1 (Mortensen, Rapp & Båmstedt, 1998; Mortensen, 2001; Roberts, 2002; Sabatier et al., 2012; Lartaud et al., 2013).

Extension rates in almost all the studies were determined only of young and newly developed corallites (Mortensen, 2001; Brooke & Young, 2009), while older polyps grew with a rate of 1.3 ± 1.5 mm yr−1 as specifically assessed by Lartaud et al. (2013). Taking only newly grown polyps’ extension rates reveals also in the present study 34% higher extensions of the inshore corals in Nord-Leksa compared to all stained corallites and even 52% higher extensions compared to older polyps only. Albeit variability in growth rates may be a result of differences in methodological approaches, environmental conditions such as temperature, food supply, turbidity, hydrography, and ocean chemistry are known to control the distribution of cold-water corals and likely also influence the growth performance (Cairns & Parker, 1992; Guinotte et al., 2006; Thiem et al., 2006; White, 2007; Roberts et al., 2009; Georgian et al., 2016). The drivers of ecosystem performance and how future changes will affect different populations of L. pertusa and other cold-water coral bioherms are still not understood and need further investigation through more in situ studies, in particular (Georgian et al., 2016).

Directly obtained in situ average linear extensions of all stained corallites on several branches in the present study were in the same order of magnitude, but slightly lower than extension rates reported for L. pertusa from the northern Gulf of Mexico, likewise measured in situ via the Alizarin Red staining technique (Brooke & Young, 2009). In their experiment, average extension rates of 3.77 mm yr−1 were measured in corals deployed at an area with a high coral density (placed into coral thickets), while lower extension rates of 2.44 mm yr−1 on average were measured of corals placed in a usually non-inhabited area ~0.25 km away from the coral area after staining. The off-reef corals in the present study were placed less than 100 m away from the reefs and no difference was observed in average extension between the on- and off-reef locations in Nord-Leksa. However, the offshore corals from Sula Reef showed lower values of linear extension than the fjord corals with 44% higher extension rates inshore compared to offshore over the year. This finding is supported by the measured growth rates in weight gain, which even showed twice as high calcification rates in the Leksa locations, although not significant. The observed discrepancy in the differences among sites between the methods is most likely attributable to the allometric growth of Lophelia, which is reflected more pronounced in the assessment of linear extensions.

While the offshore corals constitute predominantly extending colonies with thick elongated corallites, inshore colonies grow rather compact in bushy branches with shorter corallites. Fragments that were size-wise similar to the Leksa fragments had less polyps/corallites and a significantly smaller volume and surface area in Sula. The differences in morphology potentially reflect the different environmental conditions the corals are exposed to, with the inshore corals being more compact due to stronger currents, while corals in the Sula Reef can expand their corallites without enhanced risk of breakage. Freiwald, Heinrich & Pätzold (1997) highlighted that Lophelia growth forms and presumably also rates vary largely with environmental factors based on growth rhythmicity observations and stable isotope data. Hence, growth rates might vary between in- and offshore reef habitats due to physicochemical environmental factors such as current velocity, oxygen saturation, light, trace elements, and food availability although the measured physical seawater parameters and carbonate chemistry parameters did not vary considerably between locations (Table 4). However, those parameters were instant samplings at the time when coral cages were recovered. For comparison of environmental variability between sites, including seasonal changes, environmental conditions should be monitored over a yearly cycle.

Comparing growth of living corals among locations revealed no significant differences between in- and offshore reef sites or between on-reef/off-reef deployments, nor between white and orange colonies. Only when all live coral fragments (white and orange) at each location are pooled, results show a statistically significant difference between both ‘off-reef’ sites in the reefs near Nord-Leksa and Sula. The higher growth rates at the inshore off-reef site would also apply to the on-reef growth in Leksa if the rates were not so variable among all specimens, since these corals exhibited even higher mean values. Thus, inshore growth rates were by trend higher than in the Sula Reef. The 44–50% (extension—calcification) lower growth rates in Sula on average may be explained by different deployment conditions, since these coral cages were placed on relatively soft sediment ground approximately 50–100 m apart from the nearest reef structures, while the off-reef corals at Nord-Leksa were situated on rather hard bottom, which is attributable to the different habitat situations at the in- and offshore reef sites. Thus, lower growth rates of the offshore corals might be a result of an inconvenient place the coral baskets were deployed at, which might have been too far away from the reef, potentially too unsheltered with regard to sedimentation and other environmental disturbances. However, looking at the fragments in the aftermath, we observed more pronounced epibiont growth and sedimentation on the Leksa corals than at Sula and also the polyp mortality was considerably higher in both fjord deployment sites. In order to validate the trend of lower growth at the outer shelf compared to the fjord conditions, this experiment should be repeated in Sula (or comparable offshore reef) with corals being deployed closer to or directly into the reef using bigger fragments of live corals with comparable volume like the fjord corals in order to enhance comparability to the inshore experimental conditions.

White vs. orange colourmorphs

Different phenotypes of L. pertusa of highly pigmented orange corals vs. the typical white appearance were assessed. Growth rates expressed in weight gain showed a greater range in orange corals compared to the white colourmorph. While white corals had maximum growth rates of 0.019% d−1, the orange corals’ growth rates reached up to 0.049% d−1, that is, more than twice as high in maximum growth rates. However, the average growth rate of orange corals was only ~30% higher and not significantly different from the white corals. While this was also true in terms of average extension rates when taking means over all locations (18% increased length growth in orange compared to white specimens), the overall range of average extensions was relatively similar in white and orange corals. So far, only few studies took a closer look into physiological differences between the colourmorphs of L. pertusa and the physiological advantages and potential costs of enhanced carotenoid concentrations in cold-water corals remain unknown and require further investigation. Neulinger et al. (2008) hypothesised nutritional differences of different colourmorphs by specific selection of certain bacterial consortia associated only to orange or white corals. While orange corals host specific gammaproteobacteria, which might utilise reduced sulphur compounds, white L. pertusa showed a dominance of highly productive Rhodobacteracea, which can exploit even small amounts of organic material as carbon sources and which may support the nutrition of L. pertusa in environments with moderate carbon supply (Neulinger et al., 2008). Thus, the white phenotype may be able to inhabit deeper waters than orange corals. Here, the differences observed between groups were related to the different sites and environmental conditions rather than the two colourmorphs. With regard to mortality, however, the orange corals showed significantly less dead polyps than white corals. Hence, orange specimens seem to be more resistant to either the handling stress or environmental influences such as sedimentation. Together with the observed trend towards higher growth rates of orange corals in situ, the results of the present study suggest that the orange colourmorph is more resilient in the inshore reef at Nord-Leksa. If this is the case and applies also to environmental changes, the orange phenotype may also be more resilient in the future with regard to ocean change. But more physiological parameters, such as the metabolic rates, need to be investigated to assess whether higher pigmentation gives rise to any kind of physiological advantage or higher stress resistance of the orange colourmorph of L. pertusa. At least with regard to framework formation, colour variation does not seem to have adverse impacts on the reef development, as it was recently shown that self-recognition even between genetically distinct colonies of these two colour variants can lead to skeletal fusion (Hennige et al., 2014b). The similar skeletal densities of white and orange corals as observed here thereby probably facilitate skeletal fusion. No matter if simple overgrowth or allogeneic tissue fusion, the ability of L. pertusa to self-recognise at a species level regardless of the phenotype or genotype helps to reduce aggression-related energetic expenditure and supports cold-water coral reefs to represent significant ecosystem engineers of the deep sea (Hennige et al., 2014b). Nevertheless, the physiological mechanisms that lead to the higher robustness of orange colonies and the relevance of pigmentation for future reef development warrant further investigation.

Mortality

Compared to recent laboratory studies with L. pertusa specimens from the same sites (Büscher, Form & Riebesell, 2017), polyp mortality in this in situ experiment was relatively high with 10–30% on average, depending on location. While the staining approach might have contributed to the mortality, this was not observed during the recovery days in the tanks on board directly after staining. Moreover, growth studies on L. pertusa using Alizarin as staining method did not show enhanced mortality over prolonged experiments and no lethal effects were detected related to the dye (Brooke & Young, 2009; Form & Riebesell, 2012; Lartaud et al., 2017).

A comparison of different sites revealed no significant differences in mortality, but a general trend towards lower mortality in off-reef sites with lowest average percentage of dead polyps in Sula in white and orange corals (65–67% lower average percentage of dead polyps than at the Leksa sites). Although levels of sedimentation were expected to be highest in Sula, where the coral baskets were placed in relatively soft sediment, sediment particles as well as overgrowth by epibionts were more pronounced on Leksa specimens and less on Sula corals. L. pertusa was found to be fairly resilient to sediment loading, since this species efficiently cleans itself through ciliary action and mucus shedding, and its survival is at risk only when completely buried for several days (Brooke, Holmes & Young, 2009; Larsson & Purser, 2011). Moreover, Brooke, Holmes & Young (2009) tested two different morphotypes of L. pertusa, the heavily calcified form with thick branches and the more fragile form with smaller branches and corallites, with regard to their tolerance towards sedimentation and burial and found no difference between morphotypes. Thus, differences in mortality among locations due to sedimentation in the different habitats and sites or due to the different morphotypes between inshore and offshore branches are rather unlikely.

Instead, the higher mortality at the inshore sites may be related to stronger environmental fluctuations with regard to abiotic factors such as temperature, salinity, and currents, as the hydrodynamic conditions inshore are more variable than offshore. Moreover, elevated concentrations of nutrients or pollutants due to aquacultures in the Trondheimsfjord, for example, may be a possible explanation for higher polyp mortality inshore. Higher nutrient levels inshore must yet be verified.

Orange specimens showed far lower mortality (less than a third) at all locations compared to white specimens. In addition to slightly higher growth rates (18–30%), this implies that another underlying mechanism than environmental differences or handling stress causes coral mortality to a different extent in the two colourmorphs.

Bioerosion and accretion of dead coral framework

Bioerosion rates

Although bioerosion contributes globally to a greater extent to reef degradation in marine habitats than physical erosion or passive chemical dissolution, this process is often being neglected in studies investigating coral growth and ocean change (Schönberg et al., 2017). Since ocean acidification is suspected to accelerate bioerosion, as experimentally demonstrated for chemical bioerosion by phototrophic microbial euendoliths and bioeroding sponges (Tribollet, Atkinson & Langdon, 2006; Tribollet et al., 2009; Wisshak et al., 2012, 2013, 2014; Reyes-Nivia et al., 2013), CaCO3 degradation by bioeroders is an essential parameter to consider with regard to the impacts of ocean change on coral reef ecosystems. In this context, the in situ bioerosion rates reported herein may serve as base-line estimation for carbonate budget modelling assessments of Norwegian cold-water coral reefs.

A statistically highly significant difference was found between both off-reef placements and the Leksa on-reef location despite mixed bioeroded material in Sula, with the on-reef fragments exhibiting four times higher bioerosion rates. This may reflect a higher abundance and rate of colonisation of bioeroders within the live reef structures and/or it may mirror environmental conditions to be more favourable for bioeroders in that reef zone. The latter is supported by the fact that the majority of macroborers in Lophelia skeletons (excavating sponges, bryozoans, and polychaetes) are filter feeders that profit from the enhanced current regime and higher food availability in the more exposed live zone of the reef. Bioerosion rates in the Sula Reef site and the Leksa off-reef location, in contrast, were not significantly different. However, this might be a result of the mixture of Sula and Leksa dead coral material in the Sula location with relatively young bioeroded fragments from Sula. With more advanced bioerosion stages from Sula and cage placements directly into the reef structures as in Nord-Leksa, bioerosion rates could perhaps be higher in the Sula Reef.

Previously, the only experimentally determined bioerosion rates from a Lophelia reef environment were those obtained via a settlement experiment carried out at the Säcken Reef in the Swedish Kosterfjord (Wisshak, 2006), where pristine limestone tiles were subjected to bioerosion for up to 2 years of exposure. During that experiment, the gravimetrically determined bioerosion rates for the 1 year platforms were quantified as −14 ± 13 g m−2 yr−1 and were thus quite similar to the overall average of −12.37 ± 9.40 g m−2 yr−1, but lower than the Leksa on-reef bioerosion rates of −23.20 ± 7.87 g m−2 yr−1 measured in the present study. However, the Säcken Reef bioerosion rates were obtained from pristine substrates, as opposed to an established bioeroder community like in the present study. Moreover, those results were highly variable and perhaps of higher methodological uncertainty, since differences in weights were very small and only little breakages through handling could have had an effect on the changes in weight, leading to potential overestimation in bioerosion rates. The bioerosion rates reported herein, in turn, need to be considered rather as a conservative estimation, since handling stress during sampling, removal of calcareous epibionts, and re-deployment may have negatively influenced or even killed part of the established bioeroders, such as the abundant bioeroding sponges that are relatively sensitive to such disturbances (own observation). Other methodological bias, including the removal of calcifying epibionts prior to and after the experiment, trapped air bubbles during buoyant weighing sessions, breakage during deployment or recovery, all potentially affect weighing results, but may also balance each other to some extent. With no possibility to accurately quantify these factors, we have to assume that the overall bias is reasonably low and the calculated in situ bioerosion rates are the closest achievable approximation of a conservative estimate for total cold-water coral community bioerosion rates.

In any case, bioerosion rates within L. pertusa from the present study are considerably lower than rates determined from the analysis of coral samples or in situ settlement experiments of shallow-water tropical coral reefs around the globe (see Wisshak, 2006 for a review). Bioerosion rates from shallow-water reefs commonly surpass 1,000 g m−2 (planar substrate surface) yr−1, which means they are about two orders of magnitude higher (see Kiene & Hutchings, 1994; Peyrot-Clausade et al., 1995; Chazottes, Le Campion-Alsumard & Peyrot-Clausade, 1995; Reaka-Kudla, Feingold & Glynn, 1996; Tribollet et al., 2002 for examples). In addition to the fact that investigations of tropical reef substrates likely have a higher planar substrate surface available to bioerosion depending on the habitat complexity, this marked difference is a reflection of a general decrease in bioerosion rates with increasing water depths and higher latitudes. Decreasing bioerosion rates with depth primarily results from the depletion of photosynthetic microendoliths and grazers feeding upon them. Decreasing bioerosion rates towards higher latitudes again results from a temperature- and light-dependant depletion of phototrophic microborers as well as the lack of grazing parrot-fish as the most effective bioeroders in tropical seas (Wisshak et al., 2010, 2011).

To date, no experimental data are available testing the effects of ocean acidification and warming on cold-water coral bioerosion. Experiments with the demosponge Cliona celata in the cold-temperate North Sea (Wisshak et al., 2014) together with experimental evidence of increasing sponge bioerosion in tropical systems (Wisshak et al., 2012, 2013; Schönberg et al., 2017 for a review) suggests that the observed feedback to ocean acidification likely applies across species and latitudes. There is no corresponding experimental data for marine fungi available to date, but data on other chemically acting microborers indicate that an increase of their bioerosion rate might apply to most bioeroders that actively lower the local pH in order to dissolve carbonate substrates (Schönberg et al., 2017). This suggests that not only sponge bioerosion is likely to increase in Lophelia reef environments, but also bioerosion by fungal and other microendoliths. More specific experimental evidence is needed to verify this hypothesis.

Carbonate accretion

The rates of carbonate accretion by calcifying epibionts determined in the present in situ experiment were surprisingly high and sometimes even compensated for bioerosion. Accretion rates of epibionts on the dead coral framework averaged 0.003% d−1, comprising one fourth of the rate of live coral growth. The higher rates of calcareous accretion in the dead coral framework compared to the counteracting bioerosion processes were possibly caused by unrealistically fast settlement on ‘pristine’ dead coral material due to the removal of calcifying organisms prior to its deployment, which may have led to an increased rate of resettlement of the available substrate.

The observed accretion of calcifying organisms on the live corals was likewise unusually high for living corals, which might have resulted from handling of the corals prior to re-deployment. Sampling and/or staining might have led to loss of some parts of the coenosarc, the corals’ outer epithelium, resulting in bare skeleton areas suitable for settlement of other organisms. Thus, overgrowth, mainly consisting of shells from Delectopecten sp., might have been facilitated by unusually ‘free’ skeleton/substrate. In terms of measured accretion rates the abundance of Delectopecten shells was, however, negligible, but was taken into account in growth rate calculations of live coral growth.

Conversion factors

In our study we compared two established methods (buoyant weighing and linear extension measurements) to directly assess natural growth rates of living corals and measured different parameters to describe the coral fragments physically (e.g. weight, volume, polyp count). These different approaches allowed for computing conversion factors of the corresponding parameters to transform growth estimates based on buoyant weight measurements to linear extension rates and vice versa, for example, and to convert standardisation parameters such as dry weight, buoyant weight, volume, and surface area of L. pertusa. This might be helpful in future studies for a better comparability of different normalisations of physiological data of this species and for broadening assumptions to non-measured parameters.

Conclusions

The present in situ growth study revealed subtle to distinct differences in morphometry, colour phenotype, growth, and bioerosion between populations of L. pertusa from different environmental settings. In situ community bioerosion rates were significantly higher in the live reef structures compared to the two off-reef sites (in- and offshore), which is consistent with the greater presence of bioeroders within the reef. With regard to calcification of living corals, specimens from both inshore deployment sites at Nord-Leksa performed better than the ones from the offshore Sula Reef. Besides, inshore corals showed a broader range of net accretion and bioerosion rates between fragments than offshore, which might be attributable to a higher genetic variation in the fjord. Being accustomed to a higher variability in environmental conditions, fjord reefs may be more resilient with regard to environmental changes, particularly if genetic diversity supports adaptive responses to future ocean change (compare Kurman et al., 2017). Orange specimens showed tendentially higher CaCO3 precipitation and a generally broader range of net growth rates, as well as significantly lower polyp mortality. This may indicate a higher stress-resistance of the higher pigmented corals, which could become prevalent for L. pertusa reefs in the future with regard to environmental stressors induced by climate change.

The present study provides first net accretion rates of live corals as well as net erosion rates of dead coral framework as first assessment of these two opposing processes in a cold-water coral reef. Results indicate overall net accretion at all studied reef sites when directly comparing the rates of both processes. However, to comparatively determine the balance between net accretion and erosion on the ecosystem scale, compatible proportions of live and dead coral framework (integrating coverage and organism abundance) need to be assessed in future studies in order to quantify the relative contributions of both processes in a census-based approach (Reef Budget; Perry et al., 2012). Reef budget analyses of reefs from various locations with differing environmental conditions help to determine the present carbonate production states of cold-water coral reefs and to understand the ecological drivers that influence reef growth dynamics. In conjunction with net growth rate estimates under future ocean conditions, reef budgets would further allow more precise assumptions about future reef resilience of cold-water coral ecosystems.

Supplemental Information

Supplemental Information 1 Raw data of all individuals (replicates) of live and dead L. pertusa fragments in a 1 year in situ experiment.

Each data point indicates the average performance of live L. pertusa with regard to net calcification, linear extension and mortality over 1 year deployment in the natural environment (Table 1). Dead coral fragments are listed with regard to their bioerosion and accretion rates over the 1 year in situ experiment (Table 2). For both sample types, buoyant and dry weight at the end of the experiment as well as area and volume of the individual replicates are presented (Tables 1 and 2). From these parameters, the conversion different factors (Table 5) are calculated.

Click here for additional data file.

The captain and crew of RV POSEIDON are greatly thanked for support during the research cruises POS455 and POS473 in 2013 and 2014, respectively. Our colleagues from the Institute of Marine Research (IMR) in Bergen, Norway, as well as the Norwegian Environment Agency (Miljø Direktoratet) and the German Federal Agency for Nature Conservation (BfN) are acknowledged for support in the regulations of export and import permits for the specimens following the Convention on International Trade in Endangered Species of Wild Fauna and Flora (CITES). Prof. Dr. Arne-Jörn Lemke and Christian Timann are thanked for performing the CT scans and their support during the measurements. The two anonymous reviewers and Carlos E. Gómez are thanked for their constructive comments on an earlier version of this manuscript.

Additional Information and Declarations

Competing Interests

Author Contributions

Field Study Permissions

Data Availability

The authors declare that they have no competing interests.

Janina V. Büscher conceived and designed the experiments, performed the experiments, analysed the data, contributed reagents/materials/analysis tools, prepared figures and/or tables, authored or reviewed drafts of the paper, approved the final draft.

Max Wisshak conceived and designed the experiments, performed the experiments, analysed the data, contributed reagents/materials/analysis tools, prepared figures and/or tables, authored or reviewed drafts of the paper, approved the final draft.

Armin U. Form conceived and designed the experiments, performed the experiments, analysed the data, contributed reagents/materials/analysis tools, authored or reviewed drafts of the paper, approved the final draft.

Jürgen Titschack analysed the data, contributed reagents/materials/analysis tools, approved the final draft.

Kerstin Nachtigall performed the experiments, contributed reagents/materials/analysis tools, approved the final draft.

Ulf Riebesell contributed reagents/materials/analysis tools, authored or reviewed drafts of the paper, approved the final draft.

The following information was supplied relating to field study approvals (i.e., approving body and any reference numbers):

Sampling was permitted by the Norwegian Directorate of Fisheries (Fiskeridirektoratet) under permit numbers 12/17918 (in 2013) and 14/1781 (in 2014).

Export and import permits for the specimens following the Convention on International Trade in Endangered Species of Wild Fauna and Flora (CITES) were admitted by the Norwegian Environment Agency (Miljø Direktoratet) and the German Federal Agency for Nature Conservation (BfN).

The following information was supplied regarding data availability:

The raw data is available at PANGEA: Büscher, Janina; Wisshak, Max; Form, Armin; Tischack, J; Nachtigall, Kerstin; Riebesell, Ulf (2019): In Situ Growth and Bioerosion Rates of Lophelia pertusa in a Norwegian Fjord and Open Shelf Cold-water Coral Habitat. PANGAEA, DOI 10.1594/PANGAEA.903093

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
