# Peer review of "In situ growth and bioerosion rates of Lophelia pertusa in a Norwegian fjord and open shelf cold-water coral habitat"

_PeerJ, doi:10.7717/peerj.7586_

## Round 0.1 · original submission · Minor Revisions

· Academic Editor

Minor Revisions

We have received three reviews and all were positive. Two recommended minor revisions and the third recommended major revisions. My reading of the latter is that they also fall under minor revisions. Please read and respond to all comments made by the three reviewers. Several remarked on some English grammatical mistakes that need to be corrected as PeerJ does not provide copy editing. I look forward to seeing your revised manuscript.

Reviewer 1 ·

Basic reporting

- The paper is clearly written but there are a few areas where the English language and grammar could be improved. Below are a few suggestions:
o Line 127 – “measured under in situ conditions” should be something like “ current conditions measured in situ”
o Line 144 – “found to being” should be “found to be”
o Line 151 – “Besides” could be deleted or changed to “However”
o Line 348 – “Massy” is not a word. The authors could say “coral fragments with less mass” or “heavy”

- I believe there are too many instances of “i.e.” and “e.g.” throughout the manuscript (specifically the introduction). The writing will flow more smoothly if the authors can eliminate a few of them and write the example(s) as a sentence.

- The introduction is well written and includes relevant information and references. I do think the authors could add a bit more about projected ocean acidification and potential changes in the aragonite saturation state in the context of cold-water corals, since they are potentially the first ones to be impacted by low saturation states.
o Refs: Orr et al., 2005; Guinotte et al., 2006; Georgian et al., 2016a

Experimental design

- This study aims to address interesting and not fully explored questions around CWC calcification and bioerosion in perspective of current environmental conditions and looking to the future under climate change.

- The authors applied a variety of different methods to measuring calcification and impacts of bioerosion. I think these multiple methods strengthen the overall experimental design and impact of this study. Specifically, the CT scanning was a very innovative approach I hope to continue to see used in similar studies.

- I am hesitant about the mixing of the Sula and Nord-Leska dead erect coral framework. The author states that the structural framework is quite different between the two sites so it would have been interesting to compare the effects of bioerosion on the different skeletal “types”.

- The authors do acknowledge the missing treatment, in that there was not an experiment run on the reef of Sula. I am curious, why did the authors choose to run the samples they did off of the reef? Vs. having two on the reef treatments (one at Sula and one at Leska)?

- I would appreciate it if the authors could discuss why they chose to use Alizarin Red S instead of another stain like Calcein. Also, do the authors think the long exposure to Alizarin Red could have contributed to some of the mortality observed?

Validity of the findings

- No need to state “We provide the first” in line 485

- While the findings of this study are very interesting, I do feel like the authors engaged in a lot of speculation throughout the discussion.
o Lines 501-504
o Lines 518-519
o Lines 651-668
o Lines 668-671
o Lines 694-696
o Lines 758-760

- I think the authors should be cautions in the section at the beginning of the discussion from lines 491-501 where they directly combine the calcification and bioerosion estimates. They mention that the “they are to be regarded as rough approximation of relative rates, assuming similar amounts of live and dead coral framework and do not reflect the actual reef carbonate budget.” However, even with this statement it is taking the data too far to say, “overall net production rates remained still positive.” I think to make a statement about net productivity of the reefs the authors would need to create or model estimates of the amount of live versus dead framework for these reefs.

- The paragraphs in the discussion (lines 551-631) about the different methods for measuring linear extension and results from previous studies could be greatly reduced.

Additional comments

- I think this is a very interesting paper and much needed in the realm of CWC experimental biology. While I agree with the authors about the need for more in situ experimentation, I wish they had focused a bit more on the importance of recording and understanding the environmental and oceanographic conditions in the regions where experiments are occurring. Yes, we need more experiments in natural settings, but those experiments won’t hold a lot of meaning if we can’t compare the findings across different geographic areas through the environmental conditions they are exposed to.

- If the data is available, I would suggest adding a table of environmental conditions (depth, temperature, dissolved oxygen, salinity, pH, total alkalinity, aragonite saturation, etc.) for the two different sites. I think this will provide some needed context for the similarities/differences between the three study sites.

- I commend the authors on including the process of dissolution (as well as calcification) in the discussion of cold-water coral reef formation. I believe this is a very important component to understanding this system that has be overlooked for far too long. While they do make a compelling case about the significant of bioerosion specifically, I do wish they had considered how to include chemical dissolution as well.

Reviewer 2 ·

Basic reporting

The manuscript is comprehensive, structured according to PeerJ standards and well-referenced with relevant literature. The figures are relevant, well labeled and described, and the raw data is supplied. The paper is generally well written, but the authors are not native English speakers and there are a few quirks throughout. These can be easily remedied through a review by a native English speaker. The sentence structure often tends towards the verbose, and as the paper is rather long, I recommend the authors work to make the language more concise.

Experimental design

The paper is based on original research within the scope of the journal; it has a well-defined question and clearing states the knowledge gap addressed by the research topic. The study was performed to a high technical standard and the methods are scientifically rigorous and described in sufficient detail to be replicated.

Validity of the findings

There is no control per se, as the experiment was focused on in situ observations rather than laboratory experimentation; however, the data is robust and statistically sound. Conclusions are well stated and linked to results. There is some speculation, which is generally noted as such.

Additional comments

This is a good and important paper that deserves to be published. I have made some minor comments and suggestions in the PDF document (attached). My major criticism is that the language is wordy, sometimes with very long and confusing sentences, and there are minor grammatical errors scattered throughout the document.. I recommend a thorough critique by the authors to make the language more concise and reduce some redundancies, and a review by a native English speaker to resolve some of the grammatical quirks.

Annotated reviews are not available for download in order to protect the identity of reviewers who chose to remain anonymous.

·

Basic reporting

This study measured in-situ growth and erosion rates (1 year) of an important cold-water coral, which provides a critical piece of information of these processes in the natural habitat. It is clear, and professional English was used throughout. Thereby, this study has the potential to improve the understanding of the calcification and dissolution processes in cold-water ecosystems. In general, the paper meets the standard, However, there are some concerns that need to be addressed.

Specific comments

intro
The Introduction provides a thorough literature background on the subject and generally is well written. However, it is to some context too extent and can be reduced.

Line 103: delete “suggest”

Line 112-121: this paragraph can be linked to the previous one and can be more synthesized.

As stated before, authors should provide some hypothesis regarding the experimental design. For example, in the results and discussion there is a good deal of text comparing coral growth and bioerosion between on-reef vs. off-reef, however it is not mentioned the reason for choosing this set-up, in the intro or in the methods.

Authors provided access to raw data

Experimental design

This study is an original primary research, which has not been done in cold-water ecosystems, thus is within the scope of PeerJ and will benefit the cold/deep-water community interested on these ecological processes.
The research questions are well defined based on the objectives and the methodological approach, however, there is no any hypothesis that can guide the reader about why or what was the main purpose of some of the experimental design. For example, what was the criteria to select the places? Why some experimental fragments were set on-reef vs. off-reef? Why Sula reef had only fragments set-up off-reef ad not on-reef?

The investigation was performed with high technical and ethical standard.

Methods are describe in most of the paper with sufficient detail, however, see specific comments in the attached document.

Those basic questions should be address somewhere in the intro as well as in the methods.


Specific comments:

Methods
The methods are clear and well presented. Clearly explain in detail all the experimental design, manipulation that were performed and data analysis for other study to replicate it.

Line 209-214: “This offshore location comprises a relatively constant habitat in terms of environmental parameters such as temperature, salinity, pH, and currents, while the selected inshore location, a reef near the island Nord-Leksa in the outer Trondheimsfjord (henceforth referred to as Leksa Reef), is exposed to a highly variable environment due to strong tidal and compensatory currents (Form et al., 2015 Cruise Report POS473)”
It would be great if authors include a table with the environmental variables from both sites, due to the importance of this in the whole context of the paper. I tried to find the values in the cruise report Form et al. 2015 (is in the citation) but did not find any table with the environmental conditions of the selected places.

Line 214: What is the reason for the two inshore locations? Why this design was not applied to off-shore locations?

Line 224-226: I would suggest this: “All samples were collected by means of the manned submersible JAGO (GEOMAR, 2017)”

Line 228: there is a typo before 500 L

Line 234: Could the authors be more specific about the size of the coral fragments?

Line 238: Based on what methodology did the authors choose the dye concentration? Is there any empirical evidence?

Line 252: I would suggest start the sentence with “Several fist-sized…”, and again, it would be better if the authors provide a value of the size.

Moreover, in the discussion section (lines 753-754), the authors refer to a different bioerosion stages (3 to 6), however it is never explained in the method section. Authors should add one or two sentences explaining this.

Line 262-265: Avoid repetitive use of JAGO. I think it only needs JAGO in the first sentence (line 262).

Lines 331-339: It would be great if the authors provide more description about the statistical analysis performed, i.e. type of routines run, type of variables used for the analysis (dependent/independent), number of effective replicates, more explanation about the pooling procedures used for the analysis, and assumptions of the data.

Validity of the findings

In general, the experiments were conducted in a sound manner.
The data is robust, especially when the analysis is done with pooling. However, need to provide statistical validation specially to support the decision of pooling (p-value not shown, see Underwood, 1997). There are parts in the methods that are not well explained, especially the type of statistical analysis performed, and why.
Moreover, need to unify the way to present the statistical support, especially in the results. The analysis would be greatly improved if the authors include environmental data, especially because the differences in response variables are speculated to be because of differences in environmental conditions.


Specific comments:
Results

Need to provide tables with the statistical analysis, i.e. ANOVA table.

I would suggest the authors to incorporate sections 3.1.1 and 3.1.2 into one section, and more synthesized. Specially section 3.1.1 that looks detached from the paper. Moreover, this part is not taken into consideration in the discussion. I would suggest report an average size of the fragments per colormorph and per site. In this way it will look more integrated into the whole paper.

Line 347: It is not clear to me why to perform post-hoc analysis for the ANOVA if only two location are compared?

Line 350: Why two different analysis for the same comparison? It is t-test? Or ANOVA? Also please unify the way you want to present the statistical support.

Line 363: This is interesting. Any hypothesis why this can be?

Line 379: The figure 4 shows P=0.003. Moreover, please unify the way you want to present the statistical support throughout the manuscript

Line 399: Please change the N value in the Fig. 7 to match the results. I am curious if the authors perform a correction for the uneven sample size, N=14 vs. N=4.
Line 439: As mentioned in the methods, I would encourage the authors to provide environmental data since it is an important part of the hypothesis and the whole analysis of the paper.

Line 449-450: I would replace the term “statistically highly significantly different” to “statistically different” and again, it is not clear to me why to use post-hoc with only two group comparisons. Please check.

Line 472-481: This is not a part of results. Suggest move to methods section.


Discussion

Line 514: It is a little bit counter intuitive to say, “cold-water coral degradation may be particularly affected”. I would say that coral degradation is enhanced by….

Line 587-591: I would suggest link the two sentences.
“Highest extensions in aquarium cultivations were obtained by Orejas et al. (2008) in L. pertusa from the Mediterranean Sea, with measured linear extensions of 15 – 17 mm yr-1 at temperatures around 12 °C”

Line 591: Reference citation from line 592 should be changed to line 591, “Orejas et al (2008)”

Line 626: Please provide citations related to extension rates.

Lune 629: change “elevated extension” to “higher extension rates”

Line 628-631: Overuse of the words “extension rates”

Moreover, is there any explanation why corals grew more inshore sites of Nord-Leksa? This should also be addressed in the discussion.

Line 751-755: I would suggest moving to methods. It is not explained the different bioerosion stages.

Lines 808-815: I would suggest moving to intro


References:
Sone references are cited but missing from the list.

Line 76: Freiwald 2009

Line 86: should read as “Reported estimates of growth rates from in situ and laboratory conditions range from 2.4 to 35 mm per year……..” not all references provided come from in-situ studies.

Line 148: Include Provan et al (2016) in the reference list

Line 161: You should include in the reference:
Neumann, A. 1966. Observation on coastal erosion in Bermuda and mesurement of the boring rate of sponge Cliona lampa. Limnol. Oceanogr. Vol. 11(1): 92-108.
This is the first paper that coined the term Bioerosion


Conclusions are well stated and summarize the findings.

---

## Round 0.2 · accepted · Accept

· Academic Editor

Accept

Thank you for satisfactory addressing the comments raised by the reviewers. It is the opinion of the reviewers and myself that your manuscript is acceptable for publication in PeerJ. I look forward to seeing it in print.

With very best regards - Mark Benfield

Reviewer 1 ·

Basic reporting

The authors did a sufficient job of addressing the "basic reporting" comments from the first review - no further comments.

Experimental design

The authors did a sufficient job of addressing the "experimental design" comments from the first review - no further comments.

Validity of the findings

The authors did a sufficient job of addressing the "validity of the findings" comments from the first review - no further comments.

Additional comments

Thank you for spending the time to thoroughly address the comments and concerns of the first review. Congrats on a great paper!

Reviewer 2 ·

Basic reporting

No comment

Experimental design

No comment

Validity of the findings

No comment

Additional comments

The authors have addressed my comments and those of the other reviewers adequately for publication. I look forward to seeing this in print.